# A Novel Homozygous Mutation in PMFBP1 Associated with Acephalic Spermatozoa Defects

**DOI:** 10.3390/biomedicines13122882

**Published:** 2025-11-26

**Authors:** Cong Liu, Xinyue Yin, Gege Yin, Jinying Wang, Yirong Chen, Yi Zhang, Jie Li, Jin Luo

**Affiliations:** 1Center for Reproductive Medicine, Renmin Hospital of Wuhan University, Wuhan 430060, China; woshiziyoudeyulc@163.com (C.L.); zhangyi2008@whu.edu.cn (Y.Z.); 2The First Clinical College, Wuhan University, Wuhan 430072, China; 18054631625@163.com (X.Y.); 2020302091013@whu.edu.cn (G.Y.); wangjinying@whu.edu.cn (J.W.); 2023305233071@whu.edu.cn (Y.C.)

**Keywords:** male infertility, acephalic spermatozoa syndrome, PMFBP1, assisted reproductive technology

## Abstract

**Background/Objectives**: Acephalic spermatozoa syndrome (ASS) is a rare subtype of male infertility characterized by headless sperm due to defective head–tail coupling. Genetic factors are recognized as the primary etiology of ASS; however, known pathogenic mutations only explain a subset of ASS cases. Further investigations are required to elucidate the underlying genetic pathogenesis of ASS and the implications of such genetic defects for assisted reproductive technology (ART) outcomes. This study aimed to identify a novel *PMFBP1* mutation in an ASS patient; investigate the effects of the identified mutation on sperm ultrastructure and PMFBP1 protein expression/stability, and assess ART outcomes using the patient’s sperm. **Methods**: One 34-year-old infertile male with ASS was enrolled. Genetic screening was performed via whole-exome sequencing (WES), followed by Sanger sequencing for mutation validation. Sperm morphological characteristics were evaluated using Diff-Quik staining (for general morphology), transmission electron microscopy (TEM, for ultrastructural analysis), and peanut agglutinin (PNA) staining. Protein expression and stability were analyzed by Western blot and cycloheximide (CHX)/MG132 assays. ART outcomes were compared between the in vitro fertilization (IVF) cycles using the patient’s sperm and those using donor sperm. **Results**: In IVF cycles, donor sperm achieved normal fertilization (characterized by two pronuclei [2PN] formation), whereas the patient’s sperm failed to form 2PN and leading to embryo fragmentation. Genetic sequencing identified a novel nonsense mutation in PMFBP1 (c.2641C>T), which introduced a premature stop codon and resulted in a premature protein product (p.Arg881Ter). Morphologically, this mutation led to complete sperm head–tail detachment, and abnormalities in acrosome structure and sperm head–neck junction. The absence of PMFBP1 protein in the patient’s spermatozoa was observed. The in vitro assays showed the c.2641C>T mutation induced expression of the truncated PMFBP1 protein and significantly altered PMFBP1 protein stability. **Conclusions**: The PMFBP1 c.2641C>T mutation impairs sperm head–tail adhesion, thereby contributing to the pathogenesis of ASS. This study expands the clinical mutational spectrum of PMFBP1-associated male infertility and provides valuable insights for the genetic diagnosis of ASS patients. Additionally, these findings may lay a foundation for the choice of therapeutic strategies targeting PMFBP1-related ASS.

## 1. Introduction

Infertility affects approximately 15% of couples worldwide, with male factors contributing to almost half of these cases [1]. Among the diverse etiologies of male infertility, morphological abnormalities of spermatozoa, such as teratozoospermia, play a significant role [2]. Acephalic spermatozoa syndrome (ASS) is a rare yet severe manifestation of teratozoospermia characterized by the abnormal head-middle ultrastructure, which is divided into type I and type II acephalic spermatozoa based on the residual proximal and distal centriole [3,4]. The consequence of this condition is often male infertility [5]. Thus, the exploration of ASS has emerged as a central focus of clinical and scientific research.

ASS is mainly caused by genetic factors [6], and with the genetic sequencing technologies, researchers have identified many mutations associated with ASS, including *SUN5*, *SPATA20*, *TSGA10*, *BRDT*, and *PMFBP1* [7,8,9,10,11,12,13,14,15,16,17,18,19,20,21]. Mutations in these genes can result in a significant reduction in normally shaped sperm within the patient’s semen. The morphology of the sperm is primarily characterized by: a high prevalence of tailless sperm heads (pinhead sperm), a limited number of headless sperm tails, and sperm exhibiting abnormal head–tail junctions [4]. In patients with this variant of ASS, the tailless spermatozoa present in the semen retain their motility; however, due to the absence of the acrosomal structure in the sperm head, a normal acrosome reaction cannot occur, resulting in fertilization failure [3].

Among these ASS-associated genes, *PMFBP1*, whose protein is primarily located in the head–tail coupling apparatus (HTCA) and forms a sandwich-like structure with SUN5 and SPATA6, acts as a scaffold within the HTCA, which is essential for maintaining the structural integrity of sperm [19,21]. Notably, biallelic mutations in the *PMFBP1* gene, located on chromosome 16 [19], have been identified in 11% of ASS cases [20]. Such mutations in *PMFBP1* have been shown to interfere with head–tail interaction, leading to the production of headless sperm [21]. Based on the ultrastructural characteristics of head–neck dislocation, ASS can be categorized into three subtypes: separation between the two centrioles (type I), detachment between the nucleus and the proximal centriole (type II), and disjunction between the distal centriole and the mitochondrial sheath (type III) [22]. The ASS linked to the *PMFBP1* mutation is typically classified as type II ASS. With the aid of reproductive technology, specifically intracytoplasmic sperm injection (ICSI), it has been shown to be effective in treating patients with ASS, particularly type II [22], achieving favorable clinical outcomes in some instances [9,16,21,22,23,24,25,26].

Here, a novel homozygous nonsense mutation in *PMFBP1* (NM_031293.3, c.2641C>T p. Arg881Ter) was identified in the patient via whole-exome sequencing. The patient showed an ASS phenotype, and significant fragmentation of embryos was observed in the process of ART treatment. Additionally, in vitro studies confirmed the changes in the protein stability of PMFBP1 caused by the mutation. The aim of this research is to identify the genetic defect in an ASS patient and to characterize the associated sperm defects and protein changes. In doing so, we seek to expand the mutational spectrum of PMFBP1 and provide further insights into the pathogenesis of ASS.

## 2. Materials and Methods

### 2.1. Patient Information

A 34-year-old infertile man was recruited to the Reproductive Center of the Renmin Hospital of Wuhan University in January 2024. Evaluation of the couple revealed normal karyotypes. The female partner had no detected mutations, with normal menstrual cycles and sex hormone profiles. Both individuals denied any significant past medical or family history of infertility. In addition, peripheral whole blood samples were obtained from the proband and his family members for genetic analyses. The study was approved by the Ethics Committee of Wuhan University (WDRY2022-K026; 16 February 2022), and written consent was obtained from the participants.

### 2.2. Whole Exome Sequencing (WES) and Bioinformatic Analysis

Peripheral blood samples were collected from three individuals, including the proband and his unaffected parents. Genomic DNA was extracted using a BasePure Blood DNA Kit (QIAGEN, Suzhou, China) according to the manufacturer’s instructions. Whole exome sequencing was conducted to screen and identify variants across the entire exome, and novel rare *PMFBP1* variants were identified by comparison with the reference sequence Genome Reference Consortium Human Build 37 (GRCh37). Genotyping of single-nucleotide variants and insertions/deletions (indels) was performed on recalibrated BAM files using the Genome Analysis Toolkit (GATK) version 4.0, with subsequent annotation performed via ANNOVAR (version 8 June 2020) [27] across multiple databases. These databases included population frequency data, Human Genome Variation Society (HGVS) mutation descriptions, phenotypic or disease associations, and mutation function predictions. Variants were filtered based on the following criteria: (i) exonic and splicing mutations; (ii) a minor allele frequency of less than 1% in public databases such as gnomAD [28], ExAC [29], and the 1000 Genomes Project [30]; (iii) nonsynonymous mutations; (iv) mutations exhibiting high gene expression in human testes; and (v) missense mutations predicted to be deleterious by SIFT [31], PolyPhen-2 [32], and/or Mutation Taster [33].

For multiple sequence alignment of PMFBP1 homologs, Clustal Omega (version 1.2.4) was used to analyze amino acid sequences from human, chimpanzee, horse, dog, bovine, rat, and mouse, which were retrieved from the UniProt database. Structural prediction of wild-type and mutant PMFBP1 was performed using SWISS-MODEL.

### 2.3. Sanger Sequencing of PMFBP1

The *PMFBP1* variant was confirmed by Sanger sequencing using specific primers (forward primer: 5′-GAATGTCAGTTCCTAATG-3′; reverse primer: 5′-TCAAGCTTCTGCTGCTGCTG-3′). Polymerase chain reaction (PCR) amplification was conducted with Ex Taq DNA Polymerase (TAKARA, Kyoto, Japan), followed by bidirectional sequencing performed by Sangon Biotech (Shanghai, China).

### 2.4. In Vitro Fertilization and Embryo Culture

Oocytes for the first and second cycles were retrieved following the antagonist protocols. In the first cycle using the husband’s sperm, all six metaphase II (MII) oocytes retrieved were subjected to ICSI combined with artificial oocyte activation (AOA). Specifically, one hour post-ICSI, oocytes were transferred to G-1 Plus medium (Vitrolife, Göteborg, Sweden) droplets containing 10 μmol/L ionomycin (Sigma-Aldrich, St. Louis, MO, USA) for a precise exposure of 10 min to induce AOA. Following activation, oocytes were thoroughly rinsed and returned to fresh G-1 Plus culture medium. In the second cycle using donor sperm, all seven retrieved MII oocytes were incubated with prepared sperm for conventional IVF according to the laboratory’s standard protocol.

Fertilization was assessed 16–18 h post-insemination for both cycles. All resulting embryos were cultured in a time-lapse incubator (EmbryoScope, Vitrolife, Göteborg, Sweden) at 37 °C under a triple-gas atmosphere of 6% CO_2_ and 5% O_2_. The embryos were initially cultured in G-1 Plus medium for the first three days, followed by extended culture in G-2 Plus medium (Vitrolife, Sweden) for an additional three days to assess blastocyst formation.

### 2.5. Semen Analysis

Routine semen examination was performed according to the 5th edition of the World Health Organization (WHO) guidelines. The semen analysis of the patient was repeated twice, with a one-month interval, and each analysis was conducted after 7 days of sexual abstinence. As previously described [34], semen samples were evaluated according to the following parameters were observed: semen volume, progressive motility (PR), non-progressive motility (NP), total motility, sperm concentration and total sperm count (TSC). Sperm morphology was assessed through the Diff-Quik staining, where semen samples were liquefied at 37 °C and then smeared onto glass slides. The air-dried semen smears were then fixed with Diff-Quik Fixative for approximately 20 s or until thoroughly dried. The smears were subsequently stained with Diff-Quik Stain I for 10–20 s, followed by a gentle rinsing with water to remove excess stain. Finally, the smears were stained with Diff-Quik Stain II for 5–10 s, after which they were again rinsed gently with water, followed by air-drying again. The slides were then examined under a microscope to assess sperm morphology.

### 2.6. Peanut Agglutinin (PNA) Staining

Semen samples were adjusted to an appropriate concentration. An appropriate amount of sperm suspension was dropped onto a microscope slide, evenly distributed with a gentle pipette and allowed to dry to adhere the sperm to the slide. The smears were then fixed in 4% paraformaldehyde for 20 min at room temperature, followed by three rinses in phosphate-buffered saline (PBS) for 5 min each. The smears were permeabilized with 0.1% Triton X-100 for 10 min at room temperature and then rinsed again with PBS. The smears were then blocked in 1% BSA-PBS for 30 min. After blocking, the slides were incubated with PNA staining solution (containing TRITC-conjugated PNA) for 10 min in the dark. After three washes with PBS to remove unbound staining solution, the slides were stained with 4′,6-diamidino-2-phenylindole (DAPI) for 5 min and visualized under a fluorescence microscope (Zeiss AxioImager M2, Oberkochen, Germany).

### 2.7. Transmission Electron Microscopy Analysis

Semen samples from patients and controls were centrifuged and then fixed with the EM fixative for 4 h. After washing three times with 0.1M PBS (pH 7.4) for 15 min each time, the samples were post-fixed with 1% osmium acid for 2 h at room temperature. After three more washes with PBS, the samples were sequentially dehydrated through a series of alcohol solutions with concentrations ranging from 50% to 100% for a duration of 15 min each. Afterward, the fixed spermatozoa were embedded in the epoxy resin EPON 812 (90529-77-4; SPI, West Chester, PA, USA) and then cut into 60–80 nm thick slices using a Leica EM UC7 ultramicrotome (Leica Microsystems, Wetzlar, Germany). Finally, the sections were stained with lead citrate and examined with a transmission electron microscope (Tecnai G2 20 TWIN; FEI Company, Hillsboro, OR, USA).

### 2.8. Western Blotting

In this experiment, samples of sperm or 293T cells were collected after centrifugation and the radioimmunoprecipitation assay (RIPA) buffer was added to the samples. The mixture was then placed in an ice bath for 30 min during which the cells were repeatedly bead-beaten with a pipette to ensure complete cell lysis. The resultant solution was then subjected to centrifugation at 4 °C at 13,000× *g* for 5 min, after which the clear upper layer, i.e., the total protein solution, was collected. The protein concentration was quantified with a bicinchoninic acid (BCA) assay utilizing a kit from Wuhan Servicebio Co., Ltd. (Wuhan, China). The extracted proteins underwent separation via 10% SDS-polyacrylamide gel electrophoresis (SDS-PAGE) and were subsequently transferred to polyvinylidene fluoride (PVDF) membranes. Following a 1 h blocking step at room temperature with 5% nonfat milk in Tris-buffered saline-Tween (TBST), the membranes were incubated overnight at 4 °C with the primary antibody against PMFBP1 (rabbit, Proteintech, Rosemont, IL, USA). After this incubation period, the anti-rabbit IgG secondary antibody (Proteintech, USA) was applied and allowed to incubate for 1 h at room temperature. Visualization of protein bands was achieved through the application of ECL Western blotting detection reagents (MA0186; MeilunBio, Dalian, China), and subsequent analysis was conducted using the JS-1070P chemiluminescence imaging system from Peiqing (Shanghai Peiqing Science & Technology Co., Ltd., Shanghai, China).

### 2.9. Protein Stability Analysis

The mutant and wild-type (WT) *PMFBP1* cDNAs were chemically synthesized by Sangon Biotech Company (Sangon Biotech (Shanghai) Co., Ltd., Shanghai, China) and subsequently cloned into a pcDNA3.1+C-HA vector, which could express PMFBP1 WT or mutant +HA fusion protein with the HA tagged at the C-terminus end of PMFBP1 WT. Then, the constructed plasmids were separately transfected into HEK-293T cells using Lipofectamine TM3000 reagent (Invitrogen, Carlsbad, CA, USA) for 24 h. The cells were cultured in DMEM (HyClone, Logan, UT, USA) supplemented with 10% fetal bovine serum (FBS) and 100 U/mL penicillin and streptomycin in a humidified 5% CO_2_ atmosphere. The HA-tagged *PMFBP1* WT or mutant plasmids transfected into HEK-293T cells were treated with cycloheximide (CHX; 100 μmol/L) or with MG132 (10 μmol/L) for 0, 4, 8, and 12 h. After treatment, the cells were collected and analyzed by Western blot to determine the expression level.

## 3. Results

### 3.1. IVF Outcomes

As shown in Table 1, the patient’s semen showed low concentration and impaired progressive motility. In the first cycle using the patient’s sperm, a total of six oocytes were retrieved and ICSI was performed due to poor sperm quality. In the meantime, an AOA was performed because no normally headed spermatozoa were found. Unfortunately, no normally fertilized zygotes were observed on the day of the pronuclear check, except for four oocytes with 1PN. On day 3, the 1PN embryos showed significant fragmentation, and no viable embryos were obtained. As a result, the cycle was canceled. Compared with the first cycle, donor sperm was adopted in the second cycle, which showed normal sperm parameters (Table 1). Seven oocytes were retrieved in the second cycle, and six oocytes were successfully fertilized after IVF, resulting in four viable embryos (Figure 1a, Table 1).

### 3.2. Identification of a Novel Mutation in PMFBP1

WES was conducted on the male patient and revealed a homozygous c.2641C>T (p. Arg881Ter) mutation in *PMFBP1*, in which the male patient was proposed as a strong candidate. Furthermore, Sanger sequencing confirmed that the patient’s parents carried the heterozygous mutations (Figure 1b), consistent with an autosomal recessive inheritance pattern (Figure 1c). The c.2641C>T (p. Arg881Ter) mutation in *PMFBP1* is not present in the 1000 Genomes database and is reported with an extremely low allele frequency in the gnomAD database and approximately 8.25 per million base pairs in the ExAC database. MutationTaster analysis indicates that this mutation is likely pathogenic, with a CADD score of 36.0, placing it in the top 0.1% of deleterious variants in the genome (Table 2).

### 3.3. PMFBP1 Mutation Analysis

Several variants of PMFBP1 have been identified in ASS patients (Figure 2a). The homozygous nonsense variant identified in this proband (c.2641C>T) is located within exon 17 of the PMFBP1 gene. This variant introduces a premature stop codon by replacing an arginine codon (CGA) with a termination codon (TGA) at position 881. Consequently, the Ter substitution at position 881 leads to the premature termination of PMFBP1 expression (Figure 2a,b). To assess the pathogenicity of this mutation, multiple sequence alignments were performed, indicating that the mutated protein site in this patient is highly conserved across different animal species (Figure 2c). To assess the impact of the mutation on the structure of the PMFBP1, we predicted the structures of both the wild-type and mutant proteins (Figure 2d). Truncated proteins can lead to structural alterations.

### 3.4. The Mutation of PMFBP1 Leads to Abnormal Sperm Morphology

Diff-Quik staining showed that sperm from the control group exhibited normal morphology with the head and the tail closely linked. In contrast, the majority of sperm from the patient had only a headless tail, referred to as the “pinhead sperm”, and very few had only an abnormally small head, leading to a diagnosis of ASS (Figure 3a). Further analysis by PNA staining revealed abnormalities in the acrosome structure (Figure 3b). Transmission electron microscopy (TEM) further confirmed significant structural defects at the sperm head–neck junction (Figure 3c).

### 3.5. The Mutation of PMFBP1 Leads to Altered Protein Expression

In order to assess the expression level of PMFBP1 in the patient’s sperm, Western blot analysis was performed, revealing a complete absence of the PMFBP1 protein in the patient’s sperm (Figure 4a). To explore the intrinsic properties of the mutant protein, we expressed the wild-type and truncated variants in HEK293T cells. In stark contrast to its absence in vivo, the truncated PMFBP1 protein exhibited a slower degradation rate and increased stability when challenged with the protein synthesis inhibitor cycloheximide (CHX) in this cellular system. The changes in stability of the truncated protein were further confirmed in the presence of the proteasome inhibitor MG132 (Figure 4c).

## 4. Discussion

This study identifies a novel homozygous nonsense mutation in PMFBP1 (c.2641C>T, p.Arg881Ter) in an infertile male patient with acephalic spermatozoa syndrome (ASS) who experienced assisted reproductive technology (ART) failure. This mutation leads to sperm head–tail detachment, expands the mutational spectrum of PMFBP1 and reinforces that truncating mutations in this gene are a well-established cause of such severe teratozoospermia.

The *PMFBP1* gene, located on human 16q22.2, encodes a testis-specific protein of 1007 amino acids that is localized to the sperm head–tail junction [12,20,35]. PMFBP1 interacts with SUN5 and SPATA6, forming a structural complex essential for HTCA integrity [13,14,21,36]. The “sandwich” structure formed by these proteins suggests that PMFBP1 may function as a scaffold protein, linking the HTCA to the sperm nuclear membrane. In this study, we identified a nonsense variant (c.2641C>T, p.Arg881Ter) in *PMFBP1*, which introduces a premature stop codon at position 881. Previous studies have established that the C-terminal region of PMFBP1 protein is critical for the attachment of the basal body to the sperm implantation fossa [11,21] and that Therefore, the nonsense mutation of PMFBP1 results in truncation of the C-terminal domain, may disrupt this protein complex, leading to an aberrant HTCA structure and failure to maintain the normal connection between the sperm head and tail, ultimately causing ASS. Our morphological analyses, including Diff-Quik staining and transmission electron microscopy, revealed a complete disconnection between the sperm head and tail in the patient. Collectively, the genetic finding of a truncating mutation coupled with the observed structural defects is consistent with the disruption of HTCA integrity due to PMFBP1 dysfunction, which provides a plausible explanation for the ASS phenotype in this patient.

Given the potential impact of this novel mutation on the PMFBP1 protein, we performed Western blot (WB) analysis. Notably, no PMFBP1 protein expression signal was detected in the patient’s sperm samples. In order to explore the potential mechanism, we performed in vitro assays. First, in the in vitro cell system, we successfully detected the expression of mutant PMFBP1, and its molecular weight was fully consistent with the expected size of the truncated protein generated by the p.Arg881Ter mutation. However, this result contradicted the “absence of PMFBP1 expression in the patient’s sperm”; thus, we further conducted protein stability assays to rule out the possibility that “insufficient protein stability led to undetectable expression”. Unexpectedly, the results showed that following treatment with CHX (a protein synthesis inhibitor) and MG132 (a proteasomal degradation inhibitor), the stability of mutant PMFBP1 was significantly enhanced compared to the wild-type. Given that PMFBP1 is primarily localized at the sperm head–tail junction—a region that precisely exhibits core structural abnormalities (head–tail detachment) in the patient’s sperm, we propose the absence of PMFBP1 in ejaculated sperm may result from the structural disintegration of the HTCA during spermiogenesis, leading to the loss of the entire structure harboring PMFBP1, rather than its direct proteasomal degradation. Nevertheless, the validation of the mutant PMFBP1 expression during spermatogenesis in testicular sperm was not achieved as the patient declined testicular sperm extraction (TESE).

Based on the identification of the PMFBP1 p.Arg881Ter mutation in this study, we propose future research directions to deepen the understanding of PMFBP1 function and ASS pathogenesis. Utilizing testicular tissue or germ cell models will clarify the fate of p.Arg881Ter-mutated PMFBP1 during spermatogenesis, resolving the paradox between its in vitro stability and low expression in sperm. Investigations into HTCA dynamics focusing on p.Arg881Ter-truncated PMFBP1 will elucidate how its truncated domains impact HTCA integrity. Establishing animal models harboring the PMFBP1 p.Arg881Ter mutation will not only validate the mutation’s pathogenicity but also facilitate exploration of ASS pathogenesis and therapeutic research.

In summary, the identification of a novel homozygous *PMFBP1* mutation (c.2641C>T, p.Arg881Ter) in this case of asthenoteratozoospermia syndrome (ASS) adds to the evidence supporting the genetic heterogeneity of the condition. The observed protein truncation and associated sperm abnormalities confirm the detrimental effect of such loss-of-function variants, while the clinical outcome in this patient highlights potential challenges for ART. Collectively, our research broadens the mutational spectrum of *PMFBP1* and solidifies the understanding of its essential function in male fertility.

## Figures and Tables

**Figure 1 biomedicines-13-02882-f001:**
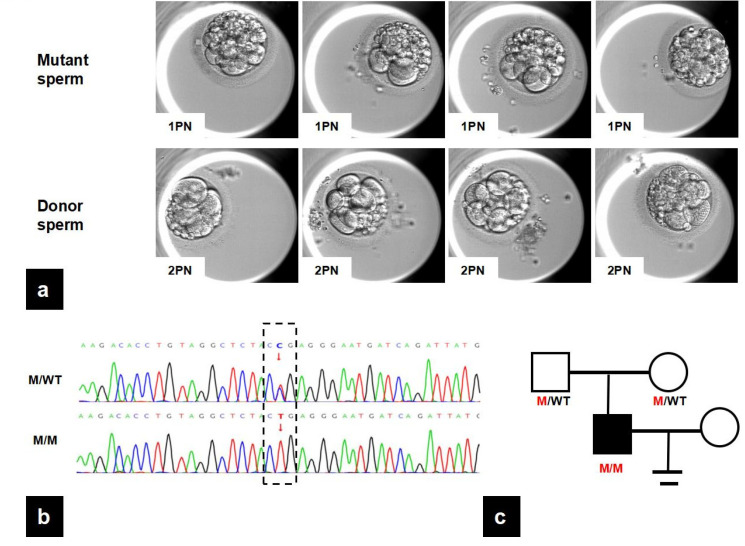
The clinic and genetic data of the proband. (**a**) Images of embryos captured at 68 h post insemination using affected sperm or donor sperm, respectively. All embryos fertilized by mutant sperm were derived from 1PN zygotes, while embryos fertilized by donor sperm were derived from 2PN zygotes. (**b**) Validation of the *PMFBP1* variant identified by WES through Sanger sequencing. The dashed box delineates the specific site of genetic variation, contrasting the mutant allele (T) identified in the patient with the conserved wild-type allele. (C) in the reference sequence. (**c**) Pedigree of the ASS family: Squares represent male family members, and circles represent female members. The mutation was validated by the parents of the proband. PMFBP1: polyamine modulated factor 1 binding protein 1; ASS: acephalic spermatozoa syndrome; M: mutation; WT: wild type.

**Figure 2 biomedicines-13-02882-f002:**
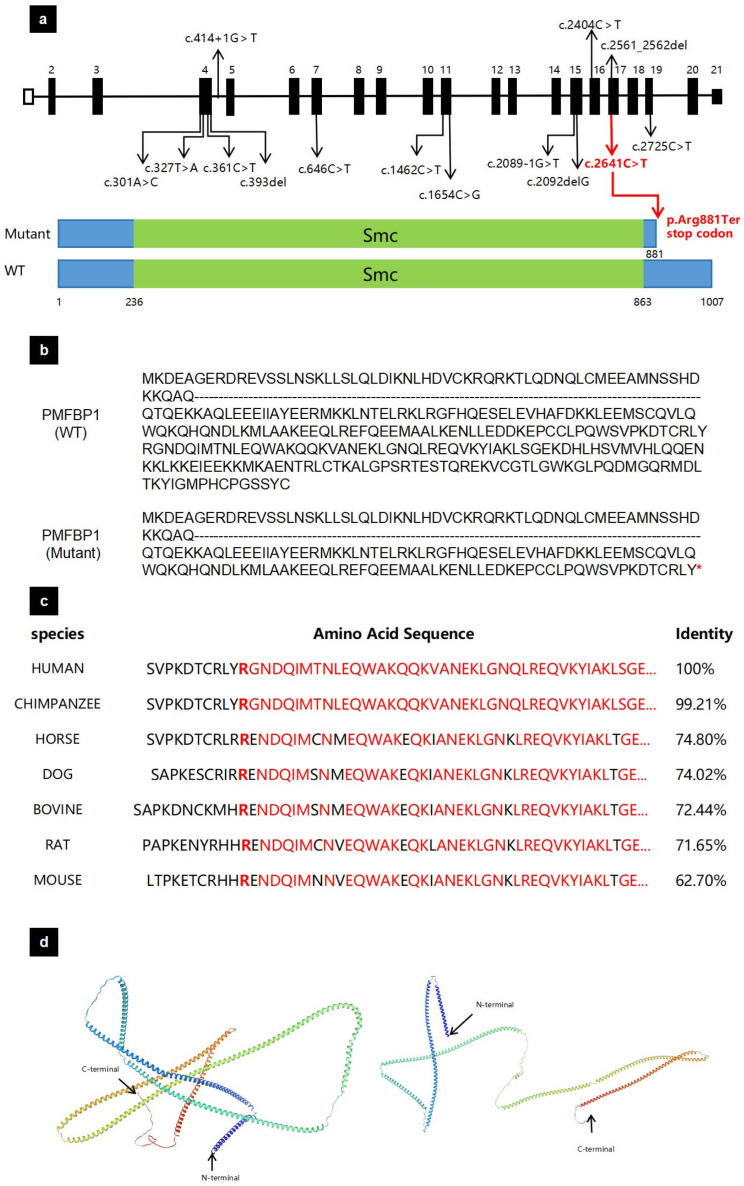
The analysis of the mutation in *PMFBP1*. (**a**) The location of identified *PMFBP1* variants and ours. Our variant is annotated in red lines in the genomic structure of *PMFBP1* and the PMFBP1 protein. Full-length PMFBP1 is 1007 AA and the Smc domain is from 236 to 863 AA (green box). The p. Arg881Ter mutation causes the protein coding to terminate at the 881st arginine. (**b**) The PMFBP1 protein variant. The asterisk (*) denotes a premature termination codon at this amino acid position, resulting in a truncated protein product. (**c**) Alignment of multiple PFMBP1 amino acid sequences across species. (**d**) Structural changes in the PFMBP1 protein based on SWISS-MODEL. PMFBP1: polyamine modulated factor 1 binding protein 1; Smc: the functional structural maintenance of chromosome region; AA: amino acid.

**Figure 3 biomedicines-13-02882-f003:**
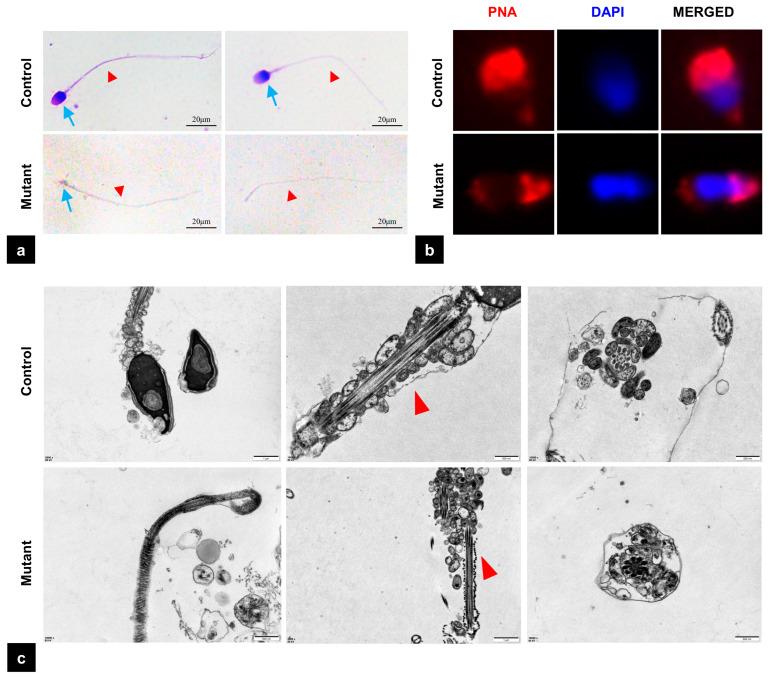
Morphological ultrastructure defects in *PMFBP1* mutant spermatozoa. (**a**) Light microscopy of Diff-Quik-stained normal sperm and the patient’s sperm. The blue arrows mark the head of the sperm, the red arrows highlight the tail of the sperm. Scale bar = 20 μm. (**b**) The acrosome and nucleus of the sperm are stained with TRITC-conjugated PNA (red) and DAPI (blue), respectively. (**c**) The TEM images of normal sperm and the patient’s sperm. The arrows indicate the disordered cervical mitochondrial sheath structures of the mutant sperm. PNA: Peanut Agglutinin; DAPI: 4′,6-diamidino-2-phenylindole; TEM: transmission electron microscopy.

**Figure 4 biomedicines-13-02882-f004:**
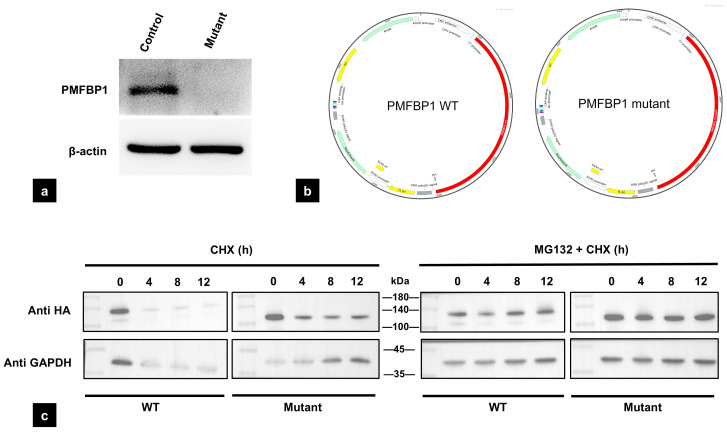
Functional analysis of the *PMFBP1* mutant. (**a**) The protein expression level decreased in the patient’s sperm after treatment. (**b**) Plasmid construction of the *PMFBP1* WT and *PMFBP1* mutant. (**c**) The stability of the protein increased after treatment with CHX (100 μmol/L) and the combination of CHX and MG132 (10 μmol/L). WT: Wild type; CHX: Cycloheximide.

**Table 1 biomedicines-13-02882-t001:** Comparative information and IVF outcomes of cycles using mutant sperm and donor sperm.

	Mutant Sperm	Donor Sperm
In vitro fertilization cycle		
oocytes retrieved	6	7
2PN fertilization rate	0	6
Utilized embryos	0	4
Semen parameters		
Concentration (×10^6^/mL)	9.6	63
Progressive motility	1	37
Total motility	9	50
Normal morphology rate	0	6

**Table 2 biomedicines-13-02882-t002:** Detailed description of the bi-allelic mutations in PMFBP1 identified in the proband.

Gene	Position	RefSeq ID	AA Alteration	Mutation Type	Status	1000Ga	Gnom ADa	ExACa	Mutation Tasterbb	CADD_ph Red
PMFBP1	Chr16:7157497	NM_031293.3	c.2641C>T p. Arg881Ter	Nonsense	Homo	NA	7.955 × 10^−6^	8.25 × 10^−6^	D	36.0

## Data Availability

The datasets used and/or analyzed during the current study are available from the corresponding author on reasonable request.

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
