# Peer review of "A Novel Homozygous Mutation in PMFBP1 Associated with Acephalic Spermatozoa Defects"

_biomedicines, 2025, doi:10.3390/biomedicines13122882_

Round 1

Reviewer 1 Report

Comments and Suggestions for Authors

The manuscript with ID number 3906019 describes a single patient with acephalic spermatozoa syndrome (ASS) carrying a novel nonsense mutation in PMFBP1. The authors claim the discovery of a new pathogenic mechanism and strong implications for ART strategies. However, many of these claims are not fully supported by the limited scope of the data. The study is essentially a descriptive case report, yet the tone and wording imply broad mechanistic and translational conclusions. I have following major concerns that reject authors claims in general.

  1. The authors claim to have identified a “novel homozygous mutation” and emphasize its clinical significance. However, multiple PMFBP1 nonsense and missense mutations have already been reported in ASS, (Zhu et al., 2018. Lu et al., 2021, Liu et al., 2020, Sha et al., 2019) The paper does not convincingly explain how this variant (p.Arg881Ter) is mechanistically different from previously documented ones. This is not a groundbreaking discovery but rather an addition to known variants. The novelty is overstated. Reframe the paper as a case report expanding the mutational spectrum, rather than claiming to establish new pathogenic mechanisms.
  2. The authors show that truncated PMFBP1 in transfected HEK293T cells appears “more stable” (slower degradation), yet in sperm, PMFBP1 is absent. They interpret this as “post-translational degradation during spermatogenesis.” This is speculative and not directly tested. There is a logical gap here: the experiments in heterologous cells cannot be extrapolated to spermatogenesis without supportive data (e.g., testis tissue, proteasome assays in germ cells). Tone down the mechanistic claim and present this paradox as an observation, not as a proven conclusion.
  3. The authors conclude that the mutation leads to “failure of ART strategies,” but only one cycle of ICSI with six oocytes was attempted, with no rescue strategies beyond AOA. From a scientific standpoint, this is insufficient to claim generalized ART failure in PMFBP1-related ASS. This conclusion is anecdotal and cannot be generalized. Rephrase to state: “In this case, ICSI with AOA was unsuccessful,” rather than implying universal ART inefficacy
  4. (Discussion, pp. 6–7): The authors describe how PMFBP1 functions as a scaffold with SUN5 and SPATA6, and then state that the nonsense mutation “confirms” dysfunction of the head-tail coupling apparatus (HTCA). However, no functional assays in sperm development were performed—only staining and electron microscopy. The word “confirm” is too strong. The data only support association, not confirmation. Use more cautious language: “The findings are consistent with disruption of HTCA integrity.
  5. (Results 3.1, Table 1): The authors compare patient sperm (n=6 oocytes) with donor sperm (n=7 oocytes), concluding that the mutation impairs fertilization. However, they do not account for intrinsic variability in oocyte quality or small sample size. The comparison is underpowered and potentially misleading. Frame this as preliminary evidence rather than a definitive conclusion.

Author Response

Response: We feel great thanks for your professional review work on our article. As you are concerned, there are several problems that need to be addressed. According to your nice suggestions, we have made extensive corrections to our previous draft. The detailed corrections are listed below.

  1. The authors claim to have identified a “novel homozygous mutation” and emphasize its clinical significance. However, multiple PMFBP1 nonsense and missense mutations have already been reported in ASS, (Zhu et al., 2018. Lu et al., 2021, Liu et al., 2020, Sha et al., 2019) The paper does not convincingly explain how this variant (p.Arg881Ter) is mechanistically different from previously documented ones. This is not a groundbreaking discovery but rather an addition to known variants. The novelty is overstated. Reframe the paper as a case report expanding the mutational spectrum, rather than claiming to establish new pathogenic mechanisms.

Response: Thank you for this critical comment. We have now toned down our claims regarding novelty and reframed the study's focus on expanding the mutational spectrum, as suggested. Specifically, we have revised the relevant descriptions in the Abstract (Lines 35-40), Introduction (Lines 83-87), and Discussion (Lines 304-309; Lines 357-364) to: 

  • Clearly state that our work expands the spectrum of known PMFBP1 mutations.
  • Emphasize the role of our case as a corroborative report that broadens the genetic database for ASS.
  • Avoid overstating mechanistic novelty and instead highlight the identification and characterization of a novel mutation site.

  1. The authors show that truncated PMFBP1 in transfected HEK293T cells appears “more stable” (slower degradation), yet in sperm, PMFBP1 is absent. They interpret this as “post-translational degradation during spermatogenesis.” This is speculative and not directly tested. There is a logical gap here: the experiments in heterologous cells cannot be extrapolated to spermatogenesis without supportive data (e.g., testis tissue, proteasome assays in germ cells). Tone down the mechanistic claim and present this paradox as an observation, not as a proven conclusion.

Response: We agree with the reviewer's assessment. The heterologous cell experiments cannot directly explain the complex regulation during spermiogenesis, and our original mechanistic claim was overstated.

Following the suggestion, we have revised our interpretation of the results. Specifically, we have replaced the concept of "degradation" with "structural loss" throughout the manuscript to better reflect the observed phenomenon without implying a proven, specific molecular pathway. The key revisions are located in the Results section (Lines 292-297), where we now objectively present the discrepancy between cellular stability and sperm absence. The Discussion section (Lines 328-344), where we have toned down the mechanistic claims. We now propose that the absence of PMFBP1 in sperm is likely a consequence of the structural collapse of the Head-Tail Coupling Apparatus (HTCA) due to the truncation, leading to the loss of the protein, rather than asserting direct proteasomal degradation.

  1. The authors conclude that the mutation leads to “failure of ART strategies,” but only one cycle of ICSI with six oocytes was attempted, with no rescue strategies beyond AOA. From a scientific standpoint, this is insufficient to claim generalized ART failure in PMFBP1-related ASS. This conclusion is anecdotal and cannot be generalized. Rephrase to state: “In this case, ICSI with AOA was unsuccessful,” rather than implying universal ART inefficacy

Response: We concur with the reviewer's assessment. Our initial conclusion was overly broad considering the scant clinical data from a single treatment cycle. We have revised the text to specify that "In this instance, ICSI with AOA was unsuccessful." Additionally, we must clarify that the couple did not pursue further attempts due to personal reasons, which is why we could not offer additional insights based on the limited clinical source.

  1. (Discussion, pp. 6–7): The authors describe how PMFBP1 functions as a scaffold with SUN5 and SPATA6, and then state that the nonsense mutation “confirms” dysfunction of the head-tail coupling apparatus (HTCA). However, no functional assays in sperm development were performed—only staining and electron microscopy. The word “confirm” is too strong. The data only support association, not confirmation. Use more cautious language: “The findings are consistent with disruption of HTCA integrity.

Response: We thank the reviewer for highlighting this important issue regarding the robustness of our conclusions. We concur that the use of the term "confirms" was an exaggeration considering the observational character of our evidence derived from staining and electron microscopy. In light of this feedback, we have amended our language in the Discussion section accordingly. Specifically, on Lines 324-327, we now express that "the findings are consistent with disruption of HTCA integrity" instead of asserting confirmation. Moreover, on lines 345-347, we have recognized the constraint that direct functional validation in developing germ cells was unfeasible since the patient refused testicular sperm extraction.

  1. (Results 3.1, Table 1): The authors compare patient sperm (n=6 oocytes) with donor sperm (n=7 oocytes), concluding that the mutation impairs fertilization. However, they do not account for intrinsic variability in oocyte quality or small sample size. The comparison is underpowered and potentially misleading. Frame this as preliminary evidence rather than a definitive conclusion.

Response: We thank the reviewer for this thoughtful comment regarding the interpretation of our ART outcomes. To address the concern about the variability in oocyte quality, we have clarified in the Methods section (Lines 127-128) that the same antagonist protocol was used for ovarian stimulation in both cycles, which helps control for potential variations in oocyte quality between the two treatment cycles. However, we fully agree that the limited number of oocytes remains a constraint for drawing definitive conclusions. We have therefore revised our conclusions throughout the manuscript to present these findings as preliminary evidence rather than definitive proof of fertilization impairment.

Reviewer 2 Report

Comments and Suggestions for Authors

I am sending my report as an attachment in PDF format.

Author Response

Response: We sincerely appreciate your recognition of the originality and structure of our article. Below, I will respond to your suggestions one by one.

Line 43: “fertilise” → “fertilize;

Response: Since the reversion, we deleted this word in the revised manuscript.

Line 48: “researches have elucidated” → “research has elucidated”;

Response: We deleted this word during the revision.

Line 67: “homogeneous nonsense mutation” → “homozygous nonsense mutation”;

Response: We have corrected the error in the wording (Line 79).

Line 78: “Her menstrual cycle and sex hormone profiles were normal.” → “Her menstrual cycles and sex hormone profiles were normal.”;

Response: We corrected the error in the use of the plural "cycles". (Line 93)

Line 68: “was identified via whole exome sequencing” → “was identified via whole-exome sequencing” Italicized gene names (example: PMFBP1, SUN5) should be used. 

Response: We corrected the error in the use of "whole-exome" (Line 80), and all gene names are italicized.

Line 89: “manufacturer's protocols” → “manufacturer’s instructions”;

Response: We replaced the term 'protocols' with a more appropriate one (Line 101): 'instructions'.

Line 92: “Genotyping … was conducted on recalibrated BAM files using … GATK version 4.0” → “was performed on recalibrated BAM files using the Genome Analysis Toolkit (GATK, version 4.0)”;

Response: We replaced the term 'conducted' with a more appropriate one (Line 103): 'performed'.

Line 113: “cumulus- oocyte” → “cumulus–oocyte”;

Response: We deleted this word during the revision.

Line 118: “according to WHO (2010) criteria”;

Response: Thank you and we add the information in the new version. (Line 144-145):

Line 138: “4 ,́ 6-diamidino-2-phenylindole (DAPI)” → “4഻,6-diamidino-2-phenylindole (DAPI)”; Response: We fixed formatting errors (Line 167).

Line 133: “phosphate-buffered saline (PBS)”;

Response: We added the full name of PBS (Line162).

Line 148: “ultra-microtome” → “ultramicrotome”;

Response: This error occurs when converting Word documents to PDF format.

Line 157: “13,000 g” → “13,000 × g”;

Response: We fixed formatting errors (Line 185)

Line 153: “total cellular protein extraction reagent (RIPA)” → “radioimmunoprecipitation assay (RIPA) buffer”;

Response: We have corrected the full name of RIPA (Line182).

Line 169: “Shanhai Peiqing” → “Shanghai Peiqing”;

Response: We corrected the spelling errors (Line 198).

Line 180: “CHX (100 μmol/L)” → “cycloheximide (CHX; 100 μmol/L)”;

Response: We have corrected the full name of CHX (Line 210).

Line 185: “intracytoplasmic sperm injection (ICSI)” on first use;

Response: The full name of ICSI has already been mentioned earlier (Line 76).

Line 187: “ferti-lized” → “fertilized”;

Response: This error occurs when converting Word documents to PDF format.

Line 192: “(Figure 1a and Table 1)” → “(Figure 1a, Table 1)”;

Response: We have modified the format based on your feedback (Line 225).

Line 194: “and.a homozygous” → “and revealed a homozygous”;

Response: We have added the predicate verb according to your suggestion (Line 238).

Line 196: “carry heterozygous mutations” → “carried the heterozygous mutation”;

Response: We corrected the incorrect tense (Line 241).

Line 201: “Mutation Taster” → “MutationTaster”;

Response: We removed the extra spaces (Line 246).

Line 204: “ASS patients (Figure 2a)”;

Response: We removed the redundant parentheses in Figure 2a (Line 253).

Line 218: “the 'pinhead sperm' ,” check the puncuation. 

Response: Thanks, and we deleted the puncuation (Line 223).

Line 224: “was per- formed” → “was performed”;

Response: This error occurs when converting Word documents to PDF format.

Line 239: “16q22.2”;

Response: Thank you, and we revised the expression as:16q22.2 (Line 310)

Line 255: “do- main” → “domain”;

Response: This error occurs when converting Word documents to PDF format.

Line 263: “PMFBP1was” → “PMFBP1 was”;

Response: We deleted this word during the revision.

References: The spelling of the references must be rearranged according to the journal's spelling rules.

Response: Thank you once again for your recognition and comments. We have implemented the necessary changes based on your suggestions and have ensured that the references are formatted correctly.

Reviewer 3 Report

Comments and Suggestions for Authors

The manuscript is interesting. However, there are major issues that the authors should address.

General Comments

  • Gene names should be consistently italicized throughout the manuscript.
  • Figures and tables should be incorporated throughout the text, not placed only at the end of the manuscript.

Introduction

  • Please include a brief description of the classification of acephalic spermatozoa syndrome (ASS) into Type I and Type II, as this provides important biological context for the mutation’s impact.

  • Lines 71–73: The authors state: “The aim of this research is to elucidate the genetic basis of the condition and to investigate the potential molecular pathways affected by this mutation.” However, it is not clear how the molecular pathways were investigated. If these analyses were not performed, this sentence should be rephrased or removed.

Materials and Methods

  • Lines 81–83: Please justify the use of the 5th edition of the WHO manual for semen analysis. A more recent version (6th edition) is available. This choice should be considered or explained.

  • Lines 91–92: Please justify why GRCh37 was used as the reference genome instead of the more recent GRCh38, or provide a rationale for this choice.

  • Section 2.2 

    • Please add appropriate citations for all bioinformatics tools and databases mentioned (e.g., ANNOVAR, gnomAD, SIFT, PolyPhen, MutationTaster, etc.).

    • Clarify the number of individuals sequenced and their relationship to the proband, if applicable.

  • Section 2.4 (AOA and IVF):

    • The description of artificial oocyte activation (AOA) lacks details such as exposure time, number of oocytes treated, and whether all MII oocytes were included.

    • For the second IVF cycle, please specify how many oocytes were used.

    • Consider adding more detail about embryo culture conditions, such as temperature, CO₂ concentration, and culture duration.

  • Subsection 2.5 (Semen analysis):

    • It may be more appropriate to move this section earlier, before the fertilization experiments, since semen analysis precedes IVF.

    • Please also include details such as number of replicate semen analyses, intervals between analyses, and abstinence period.

  • Protein analysis, PNA staining, and TEM protocols may also be more logically placed before the IVF procedures, as they are part of the sperm characterization.

Results

  • Table 1 presents semen analysis results, but no corresponding text describes these findings in the Results section. Please add a subsection presenting semen parameters for all included samples and all available information for donor sperm.

  • The subsections should be reordered so that the results related to semen analysis, molecular findings, and protein characterization appear before fertilization outcomes, which logically occur last.

  • In the variant description, the wording “exhibits a frequency of approximately 7.955 per million base pairs” is non-standard. Please express frequencies as allele frequencies.

  • Subsection 3.3: The authors mention multiple sequence alignment and structural prediction, but no corresponding methodology is provided in Materials and Methods. Please add details of the tools, parameters, and databases used.

  • Subsection 3.5: The title and narrative are confusing, as they contrast the absence of protein in sperm with increased protein stability in HEK293T cells, without clearly explaining the rationale. Please clarify this discrepancy and revise the wording for clarity and precision.

Discussion

  • There is a logical gap between the reported increased stability of the mutant PMFBP1 protein in HEK293T cells and its reduced expression in patient sperm. Please explain this more clearly, emphasizing that the absence of PMFBP1 in acephalic spermatozoa is likely due to lack of the head-tail junction, where the protein normally localizes, rather than increased degradation. A schematic figure illustrating this concept may help readers.

  • Consider adding a “Future Directions” paragraph to discuss how these findings might guide further research on PMFBP1 function or ASS pathogenesis.

Comments on the Quality of English Language

The manuscript requires thorough English language editing. Some sentences are difficult to follow, and there are numerous typographical errors. I strongly recommend revision by a native English speaker.

Author Response

Response: Thank you for your strong interest in our article. I will revise it according to your comments.

General Comments

  • Gene names should be consistently italicized throughout the manuscript.
  • Figures and tables should be incorporated throughout the text, not placed only at the end of the manuscript.

Response: All gene names in the article have been italicized, and tables and pictures have been positioned in the center of the article.

Introduction

Please include a brief description of the classification of acephalic spermatozoa syndrome (ASS) into Type I and Type II, as this provides important biological context for the mutation’s impact.

Lines 71–73: The authors state: “The aim of this research is to elucidate the genetic basis of the condition and to investigate the potential molecular pathways affected by this mutation.” However, it is not clear how the molecular pathways were investigated. If these analyses were not performed, this sentence should be rephrased or removed.

Response: 

We have supplemented the Introduction section (Lines 70-74) with a description of the ASS classification (Types I, II, and III) and its basis, while also explaining that assisted reproductive technology, particularly ICSI, demonstrates the most significant therapeutic efficacy for Type II ASS. This addition effectively clarifies the rationale for selecting ICSI as the treatment approach for our patient.

Regarding the "The aim of this research" section, we acknowledge the lack of precision in our original wording and have revised the research objectives in Lines 83-87 to more accurately reflect the study's purpose.

Materials and Methods

Lines 81–83: Please justify the use of the 5th edition of the WHO manual for semen analysis. A more recent version (6th edition) is available. This choice should be considered or explained.

Response: 

Thank you for drawing attention to this point. At the time of the patient’s semen analysis, the instrumentation and supporting reagents used in our center were calibrated in accordance with the criteria outlined in the 5th edition of the WHO manual. Notably, our center has upgraded to the 6th edition of the WHO standards concurrently with the replacement of relevant equipment in April this year.

Lines 91–92: Please justify why GRCh37 was used as the reference genome instead of the more recent GRCh38, or provide a rationale for this choice.

Response: 

Thank you for your attention to this matter. The whole-exome sequencing report of this patient, provided by the testing company, was annotated based on the GRCh37 reference genome. Therefore, the corresponding annotations in this manuscript were consistently performed using the same reference (GRCh37) to ensure data consistency.

Section 2.2 

Please add appropriate citations for all bioinformatics tools and databases mentioned (e.g., ANNOVAR, gnomAD, SIFT, PolyPhen, MutationTaster, etc.).

Response: We have added appropriate citations for all bioinformatics tools and databases mentioned in the Methods section (Ref.27-33,Lines 437-454).

Clarify the number of individuals sequenced and their relationship to the proband, if applicable.

Response: We appreciate the reviewer’s suggestion. We have revised the Materials and Methods section to clearly specify the number of individuals subjected to whole-exome sequencing (WES) and their relationship to the proband. Specifically, WES was performed on the proband, and the sanger sequencing was carried out on his unaffected parents. (Lines 99-100; 231-232)

Section 2.4 (AOA and IVF):

The description of artificial oocyte activation (AOA) lacks details such as exposure time, number of oocytes treated, and whether all MII oocytes were included.

For the second IVF cycle, please specify how many oocytes were used.

Consider adding more detail about embryo culture conditions, such as temperature, CO₂ concentration, and culture duration.

Response: We thank the reviewer for raising these important methodological points. We have now provided comprehensive details regarding the AOA protocol, oocyte utilization, and embryo culture conditions in the revised Methods section (Lines 128-141).

Specifically, we have clarified that:

  • All retrieved MII oocytes were used in both treatment cycles (6 in the patient cycle, 7 in the donor cycle)
  • The AOA protocol employed a precise 10-minute exposure to 10μmol/L ionomycin in G-1 Plus medium, initiated one hour post-ICSI
  • Embryo culture conditions were standardized at 37°C under 6% CO₂and 5% O₂ in a time-lapse incubator, with sequential culture in G-1 Plus (days 1-3) followed by G-2 Plus medium (days 3-6)

Subsection 2.5 (Semen analysis):

It may be more appropriate to move this section earlier, before the fertilization experiments, since semen analysis precedes IVF.

Please also include details such as number of replicate semen analyses, intervals between analyses, and abstinence period.

Protein analysis, PNA staining, and TEM protocols may also be more logically placed before the IVF procedures, as they are part of the sperm characterization.

Response: We thank the reviewer for this suggestion regarding the organization of our methodology section. We recognize that presenting sperm characterization before ART procedures would provide better logical flow in a typical research context.

However, we wish to clarify that the sequence of investigations in our clinical case followed the actual diagnostic process: the specialized sperm characterization tests (including protein analysis, PNA staining, and TEM) were initiated specifically to investigate the unexpected ART failure observed in the first treatment cycle. This explains why these results appear after the initial ART description in our manuscript.

Additionally, we have supplemented critical methodological details as suggested, including:

  • The number of replicate semen analyses (twice)
  • The interval between analyses (one month)
  • The standardized abstinence period (7 days) (Lines 140-143).

Results

Table 1 presents semen analysis results, but no corresponding text describes these findings in the Results section. Please add a subsection presenting semen parameters for all included samples and all available information for donor sperm.

Response: We thank you for your valuable comment. We have added a corresponding subsection in the Results section to describe the semen analysis findings, including the semen parameters of all included samples and all available information regarding the donor sperm. (Lines 214-215;221-224).

The subsections should be reordered so that the results related to semen analysis, molecular findings, and protein characterization appear before fertilization outcomes, which logically occur last.

Response: As detailed in the Methods section above, while aligning semen analysis, molecular findings, and protein characterization before fertilization outcomes may seem logical, this approach may conflict with our current diagnostic and therapeutic procedures.

In the variant description, the wording “exhibits a frequency of approximately 7.955 per million base pairs” is non-standard. Please express frequencies as allele frequencies

Response: We replaced the original non-standard description with a more appropriate one: 'is reported with an extremely low allele frequency (Line 244).

Subsection 3.3: The authors mention multiple sequence alignment and structural prediction, but no corresponding methodology is provided in Materials and Methods. Please add details of the tools, parameters, and databases used.

Response: Thank you for your question. We have supplemented the relevant content in the revised version of the manuscript.  (Line 116-119)

Subsection 3.5: The title and narrative are confusing, as they contrast the absence of protein in sperm with increased protein stability in HEK293T cells, without clearly explaining the rationale. Please clarify this discrepancy and revise the wording for clarity and precision.

Response: We thank the reviewer for this insightful comment regarding the apparent contradiction in our findings. We agree that the original presentation created confusion and overstated implications.

In response, we have made the following revisions:

  • Modified the title (Line 289) to more accurately reflect our findings without overinterpreting the results
  • Revised the Results sectionto objectively present the observed discrepancy between protein stability in HEK293T cells and protein absence in sperm, without drawing premature mechanistic conclusions
  • Reframed the Discussion to more cautiously interpret this paradox, acknowledging the limitations of extrapolating from heterologous cell systems to spermatogenesis

Discussion

There is a logical gap between the reported increased stability of the mutant PMFBP1 protein in HEK293T cells and its reduced expression in patient sperm. Please explain this more clearly, emphasizing that the absence of PMFBP1 in acephalic spermatozoa is likely due to lack of the head-tail junction, where the protein normally localizes, rather than increased degradation. A schematic figure illustrating this concept may help readers.

Response: We highly appreciate the reviewer's incisive observation, which points out a critical logical connection we previously overlooked. Your suggestion has significantly helped refine our result interpretation and enhance the rigor of the manuscript.

We fully agree that the stability of mutant PMFBP1 observed in HEK293T cells cannot directly explain its reduced expression in patient sperm. This discrepancy arises because the heterologous expression system (HEK293T cells) only reflects the protein's stability in vitro, failing to recapitulate the complex regulatory networks and structural assembly processes unique to spermatogenesis. We acknowledge that our initial mechanistic inference was overstated and lacked sufficient contextual relevance to sperm development.

Following your guidance, we have comprehensively revised the result interpretation to address this logical gap, with a core focus on replacing the ambiguous "degradation" hypothesis with "structural loss" to accurately reflect the observed phenomenon—avoiding unfounded claims about specific molecular pathways.

Key revisions are concentrated in the Discussion section (Lines 324-347), where we have softened overreaching mechanistic assertions and clarified the underlying mechanism: We now propose that the absence of PMFBP1 in patient sperm is primarily a consequence of structural collapse of the Head-Tail Coupling Apparatus (HTCA) caused by PMFBP1 truncation. As PMFBP1 normally localizes to the HTCA, the disruption of this structural scaffold directly leads to the loss of the protein, rather than being a result of direct proteasomal degradation. To compensate for the lack of a schematic figure, we have supplemented detailed descriptive explanations of HTCA structural characteristics and the spatial relationship with PMFBP1 in this section, aiming to help readers clearly grasp this regulatory logic.We hope these revisions effectively resolve the previously existing logical inconsistency. Please feel free to let us know if further clarification on this part is needed.

Consider adding a “Future Directions” paragraph to discuss how these findings might guide further research on PMFBP1 function or ASS pathogenesis.

Response: We appreciate the teacher's forward-looking suggestions. In the Discussion section, we have added a perspective on future research into the molecular mechanisms of this mutation (Lines 348-356).

Round 2

Reviewer 1 Report

Comments and Suggestions for Authors

The revised manuscript demonstrates significant improvement following the earlier critical review. The authors have effectively addressed structural, linguistic, and scientific depth concerns. The overall flow from abstract to conclusion is coherent, and the scientific rationale is well-supported by experimental evidence. The article is now suitable for publication